



# Accounting for the photochemical variation of stratospheric $NO_2$ in the SAGE III/ISS solar occultation retrieval

Kimberlee Dubé[1], Adam Bourassa[1], Daniel Zawada[1], Douglas Degenstein[1], Robert Damadeo[2], David Flittner[2], and William Randel[3]

[1]Institute of Space and Atmospheric Studies, University of Saskatchewan, Saskatchewan, Canada
[2]NASA Langley Research Center, Hampton, VA, USA
[3]National Center for Atmospheric Research, Boulder, CO, USA

**Correspondence:** Kimberlee Dubé (kimberlee.dube@usask.ca)

**Abstract.** The Stratospheric Aerosol and Gas Experiment (SAGE) III has been operating on the International Space Station (ISS) since mid 2017. Nitrogen dioxide ($NO_2$) number density profiles are routinely retrieved from SAGE III/ISS solar occultation measurements in the middle atmosphere. Although $NO_2$ density varies throughout the day due to photochemistry, the standard SAGE $NO_2$ retrieval algorithm neglects these variations along the instrument's line of sight by assuming that

the number density has a constant gradient within a given vertical layer of the atmosphere. This assumption will result in a retrieval bias for a species like $NO_2$ that changes rapidly across the terminator. In this work we account for diurnal variations in retrievals of $NO_2$ from the SAGE III/ISS measurements, and determine the impact of this algorithm improvement on the resulting $NO_2$ number densities. The diurnal correction is applied by first undoing the SAGE III/ISS retrieval using publicly available SAGE III/ISS products to obtain an optical depth profile. The retrieval is then performed with a new matrix that ap-

plies photochemical scale factors for each point along the line of sight according to the changing solar zenith angle. In general $NO_2$ that is retrieved by accounting for these diurnal variations is more than 10% lower than the standard algorithm below 30 km. This effect is greatest in winter at high latitudes, and generally greater for sunrise occultations than sunset. Comparisons with coincident profiles from the Optical Spectrograph and InfraRed Imager System (OSIRIS) show that $NO_2$ from SAGE III/ISS is generally biased high, however the agreement improves by up to 20% in the mid stratosphere when diurnal variations

are accounted for in the retrieval. We conclude that diurnal variations along the SAGE III/ISS line of sight are an important term to consider for $NO_2$ analyses at altitudes below 30 km.

## 1   Introduction

The Stratospheric Aerosol and Gas Experiment (SAGE) III on the International Space Station (ISS) uses solar occultation

to measure the attenuation of sunlight through the middle atmosphere (Cisewski et al., 2014). These measurements are used to retrieve vertical profiles of nitrogen dioxide ($NO_2$), as well as other atmospheric constituents, mainly ozone and aerosol





extinction coefficients. The SAGE III/ISS data complements that from the earlier SAGE II (McCormick, 1987) and SAGE III/Meteor-3M missions (Mauldin III et al., 1998), as well as data from limb-viewing instruments such as the Optical Spectrograph and Infrared Imager System (OSIRIS, Llewellyn et al., 2004), to provide a long-term record of $NO_2$. It is important to

have these data sets for understanding trends and variability in the stratosphere as $NO_2$ plays a role in the chemical depletion of the ozone layer. Several studies of the long-term trends and variability in $NO_2$ (Randel et al., 1999; Liley et al., 2000; Park et al., 2017; Galytska et al., 2019; Dubé et al., 2020) have shown a consistent increase in $NO_2$ and an associated decrease in $O_3$.

$NO_2$ is mainly destroyed by photolysis so the concentration, or density, of $NO_2$ that is measured depends greatly on the
position of the sun: there is a rapid decrease in $NO_2$ at sunrise and a rapid increase at sunset. During solar occultation measurements, such as those taken by SAGE III/ISS, the solar zenith angle (SZA) is 90° at the tangent point, but varies along the instrument's line of sight (LOS). The retrieved $NO_2$ profile has contributions from the full LOS, so it includes contributions from both the day and night sides of the Earth, which can have substantially different amounts of $NO_2$. The existing SAGE III/ISS retrieval neglects variations in SZA along the LOS by assuming the concentration of $NO_2$ has a constant gradient at
each altitude above the Earth's surface (SAGE III Algorithm Theoretical Basis Document, 2002). This is a source of systematic uncertainty in the retrieved $NO_2$ that has not been quantified for SAGE III/ISS. Previous studies have examined the effect of this variation in SZA on $NO_2$ retrieved from other occultation instruments and showed that ignoring it can result in a high bias, especially below 25 km (Gordley et al., 1996; Newchurch et al., 1996; Brohede et al., 2007). The purpose of this work is to account for diurnal variations along the LOS, assess the impact, and provide an improved SAGE III/ISS $NO_2$ data set
for further study. This is done by adding correction factors to the retrieval that account for variations in $NO_2$ due to changing solar zenith angle along the LOS. By comparing the $NO_2$ concentration from this improved retrieval to that from the existing SAGE III/ISS retrieval it is possible to determine the importance of the photochemical effect. The results are also compared to $NO_2$ retrievals from OSIRIS limb scattering measurements to determine how the photochemical correction changes the bias between $NO_2$ products from the two different instruments.

## 2  $NO_2$ Photochemistry

The $NO_2$ number density depends on local solar time (LST) and equivalently, SZA (Figure 1). There is a sharp decrease in the $NO_2$ concentration at sunrise as $NO_2$ is photolyzed to become NO. Throughout the daylight hours $NO_2$ and NO are in approximate equilibrium. At sunset NO production ceases, resulting in a rapid increase in $NO_2$. Overnight $NO_2$ decreases more slowly as it is converted to nitrogen-containing reservoir species. The exact shape of this diurnal cycle depends on altitude:
the $NO_2$ concentration at 40 km stays roughly constant during the day and night, while at altitudes with more $NO_2$ there is a steady increase during the day and decrease at night.

The values in Figure 1 were calculated with the the photochemical box model originally developed by Prather et al., 1992, which has been successively updated and is now often referred to as PRATMO (McLinden et al., 2000; Adams et al., 2017). PRATMO takes an input atmospheric state consisting of specified ozone, temperature, pressure, and air density profiles for a set



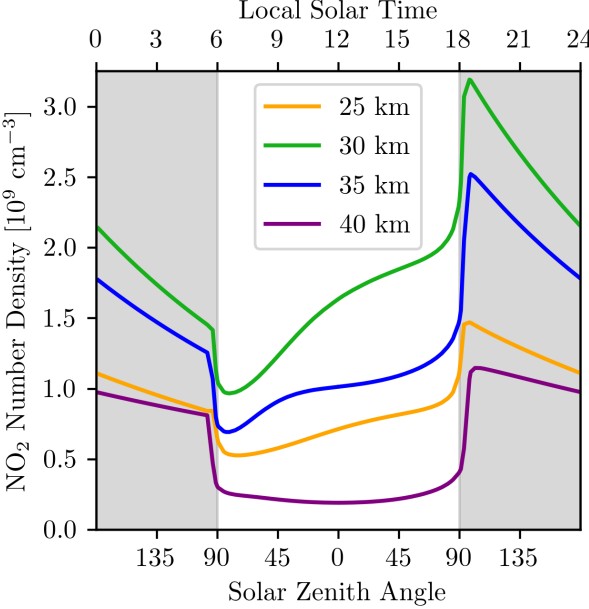

**Figure 1.** Daily cycle in $NO_2$ at the equator as a function of local solar time (top axis) and solar zenith angle (bottom axis) for four altitudes. $NO_2$ number density calculated with PRATMO photochemical box model.

latitude, longitude, and date. These input parameters are kept constant. The model then calculates a set of chemical reactions over a single day, iterating until the start and end values converge (Prather, 1992). This results in a 24 hour steady-state system of each species in the model. The model outputs are the $NO_2$ profiles at any predetermined SZAs. These values can be used to scale the measured $NO_2$ to different solar times in order to account for variations in SZA along the measurement LOS (Section 4). PRATMO scaling was used in several previous studies (e.g. Adams et al., 2017; Park et al., 2017; Dubé et al., 2020) to compare $NO_2$ from instruments that measure at different times of day.

## 3 Instruments

### 3.1 SAGE III/ISS

SAGE III has been in orbit on the ISS since March 2017. Level 2 data are available from June 2017 onwards. The ISS has an inclination of 51.6°, allowing SAGE III/ISS to view latitudes from 70°N to 70°S. The CCD spectrometer is configurable and currently observes wavelengths between ∼280nm and ∼1035nm with a resolution of 1-2nm. A separate photodiode covers ∼1542nm +/- 15nm. During each occultation SAGE III/ISS continuously scans back and forth across the sun to measure the irradiance. There are 15 sunrise and 15 sunset events per day. The coverage of SAGE III/ISS is very similar to that of SAGE II. The sunrise and sunset measurements progress in opposing directions, with each requiring about one month to achieve near global coverage.

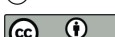



The irradiances are used in the standard SAGE III/ISS retrieval (version 5.1, SAGE III Algorithm Theoretical Basis Document, 2002) to determine the number density of several species, as well as the aerosol extinction at several wavelengths. The first step in the algorithm is to calculate slant path transmission profiles for each wavelength channel from the measured irradiance. Each slant path transmission profile is converted to a slant path optical depth profile that contains contributions from Rayleigh scattering, aerosol extinction, and absorption by at least one species. With this information $NO_2$ and $O_3$ slant

path number density profiles are solved for simultaneously using multiple linear regression. $NO_2$ is retrieved from channel S3, covering 433 to 450 nm. The slant path number density is converted to vertical number density profiles using a global fit method that assumes each layer of the atmosphere is a spherical shell with a constant gradient. The final $NO_2$ number density is available from 10 to 45 km on a 0.5 km grid with a vertical resolution of about 1.5 km. The reported uncertainty in the SAGE III/ISS $NO_2$ is around 5% at 30 km, and increases to up to 20% at 10 km and 40 km. This uncertainty is due to measurement

noise only, and does not account for systematic bias due to the horizontal homogeneity assumption.

## 3.2    OSIRIS

OSIRIS has been in sun-synchronous orbit on the Odin satellite since October 2001 (Murtagh et al., 2002; Llewellyn et al., 2004). There are 100 to 400 vertical profiles of limb-scattered solar irradiance measured each day, at wavelengths from 280 to 800 nm. $NO_2$ is retrieved by spectral fitting in the wavelength range from 435 to 477 nm for altitudes from the cloud top to

39.5 km with a resolution of 2 km.

Earlier versions of the OSIRIS $NO_2$ retrieval were developed by Haley et al. (2004), Bourassa et al. (2011), and Sioris et al. (2017). The most recent data, version 7.0, is used here for validation of SAGE III/ISS $NO_2$. Version 7.0 improves upon version 6.0 by using solar Fraunhofer lines to fit the spectral point spread function of OSIRIS rather than using pre-flight calibration values. Cloud and aerosol discrimination is also refined to better detect cloudy scenes and to push the retrieval farther into the

UTLS following the method of Rieger et al. (2019).

The OSIRIS LOS is approximately aligned with the terminator so the variation in SZA along the LOS is much smaller than for occultation instruments. McLinden et al. (2006) studied the effect of the diurnal error on $NO_2$ from OSIRIS and found that it is only significant when the SZA is near 90° and the solar azimuth angle varies significantly from 90°. These extreme conditions occurred in 16% of profiles from 2004, resulting in errors of up to 35% in the OSIRIS $NO_2$ below 25 km. Sioris et al.

(2017) used PRATMO to create a 2D OSIRIS $NO_2$ retrieval to further assess the impact of diurnal variations on the results. They found minimally improved agreement between OSIRIS $NO_2$ and $NO_2$ from balloon measurements, particularly below 20 km. Owing to the minimal effect for OSIRIS, the standard $NO_2$ data product is produced neglecting the $NO_2$ photochemical gradient.





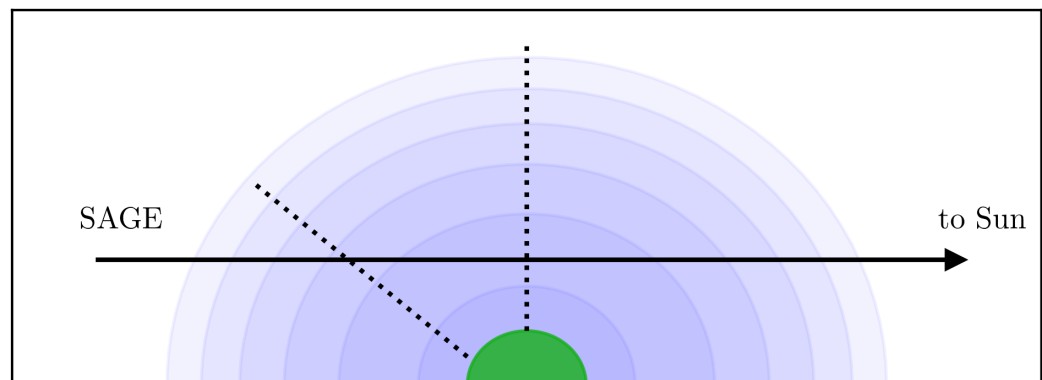

**Figure 2.** Geometry of a SAGE III occultation. The green semi-circle represents the Earth, and the blue semi-circles represent layers of the atmosphere. The angle between the dashed line and solid line is the solar zenith angle at a given location.

## 4 Retrieval

During each occultation SAGE III/ISS looks through multiple layers of the atmosphere, called shells. Each shell is defined by its altitude. Figure 2 illustrates this geometry. The black arrow represents the LOS, pointing from the instrument to the sun. The SZA at a given location is the angle between the dashed line and the LOS.

The SAGE III/ISS retrieval assumes that the number density of each chemical constituent is either constant or has a constant gradient within a shell (SAGE III Algorithm Theoretical Basis Document, 2002). This assumption is generally valid for
species such as ozone that undergo minimal diurnal variation in the stratosphere, however it is not true for $NO_2$. This can be understood by considering Figure 3. For both lines of sight in the figure the SZA at the tangent point is 90°. To retrieve the $NO_2$ concentration at the tangent point of a given LOS we need to know the $NO_2$ concentration at each shell altitude that the LOS passes through. For example, the retrieved $NO_2$ at 22 km depends on the $NO_2$ at 32 km. The $NO_2$ at 32 km is retrieved at the tangent point, where the SZA is 90°, but the 22 km line of sight passes through the 32 km shell when the SZA is around
86.8° on the near side of the tangent point and 93.2° on the far side (left panel of Figure 3). The right panel of Figure 3 shows that the $NO_2$ concentration at 32 km and the two SZAs where the 22 km LOS passes through that shell are both different from the concentration when the SZA is 90°. In addition, the $NO_2$ does not change linearly across the terminator so deviations from linearity on either side of the LOS do not cancel out. Therefore using the 32 km $NO_2$ at 90° to retrieve the 22 km $NO_2$ is inaccurate, and it cannot be assumed that the number density has a constant gradient across the terminator within a layer of the
atmosphere when performing the retrieval. This lack of spherical homogeneity can be accounted for by adding factors to the retrieval that scale the $NO_2$ according to SZA, at each location along the LOS.

Ideally we would incorporate the scale factors by redoing the conversion of slant path optical depth, obtained directly from the solar transmission measurements, to number density. As the SAGE III/ISS $NO_2$ optical depth profiles are not publicly available, we instead start by undoing the SAGE III/ISS retrieval to revert the number densities to optical depths. This is done





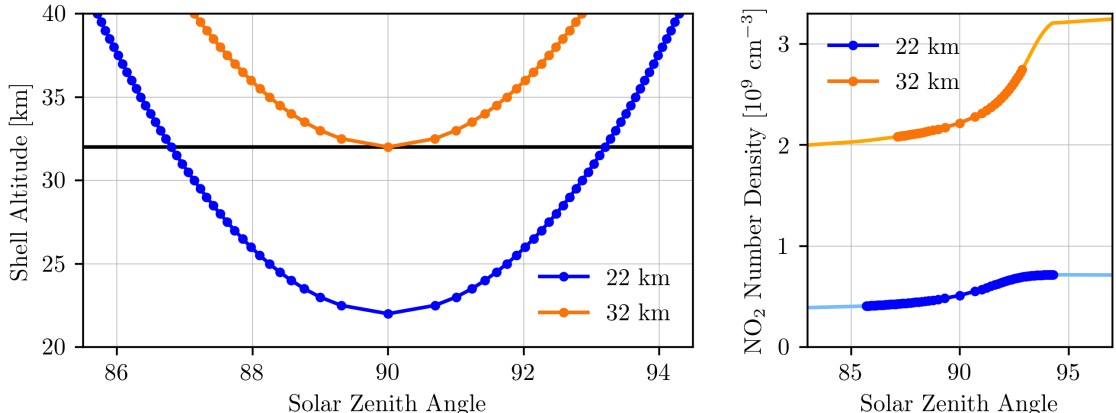

**Figure 3.** Left: Change in SZA with shell altitude along two lines of sight for a simulated sunset occultation. Right: Change in $NO_2$ at tangent altitudes of 22 km and 32 km for the same simulated sunset occultation. The dark, dotted lines correspond to the lines of sight from the left panel.

using the matrix equation,

$$\tau = \sigma X_0 n_0, \tag{1}$$

where $\tau$ is the vertical profile of slant path optical depths from $NO_2$, $\sigma$ is the $NO_2$ cross section, and $n_0$ is the number density profile. $X_0$ is the path length matrix where each row represents a LOS for a particular tangent point altitude and each column represents a different altitude through which the LOS passes. Each element of $X_0$ is the path length distance between subsequent shells along the LOS. The path lengths on opposite sides of the tangent point are the same (i.e. the distance from

shell 1 to 2 on the instrument side of the tangent point is the same as the distance from shell 1 to 2 on the sun side) which allows $X_0$ to be written as an upper triangular matrix where values from opposite sides of the tangent point are added together.

These optical depths are used to find the number densities accounting for diurnal variations, $n_{dv}$, using a new matrix, $X_{dv}$,

$$n_{dv} = \sigma^{-1} X_{dv}^{-1} \tau. \tag{2}$$

In this matrix each path length includes a factor, explained below, that depends on the SZA at that location. Note that the $NO_2$ cross section is the same in both equations 1 and 2 and so it cancels out when finding $n_{dv}$. Although this is not strictly the case, using a constant cross section is a reasonable approximation as the cross section has a weak temperature and pressure dependence. The equations also assume that optical depth is constant within each layer of the atmosphere.

For a given SAGE III/ISS scan we know the date and time, the tangent point position, the spacecraft position, and the $NO_2$

number density from the SAGE v5.1 retrieval. This information is all that is needed to construct $X_0$. To build the matrix we iterate through each LOS, moving from the tangent point to the top of the atmosphere. The LOS is the vector from the satellite to the tangent point. The effect of refraction is neglected as it is small at the altitudes being considered.





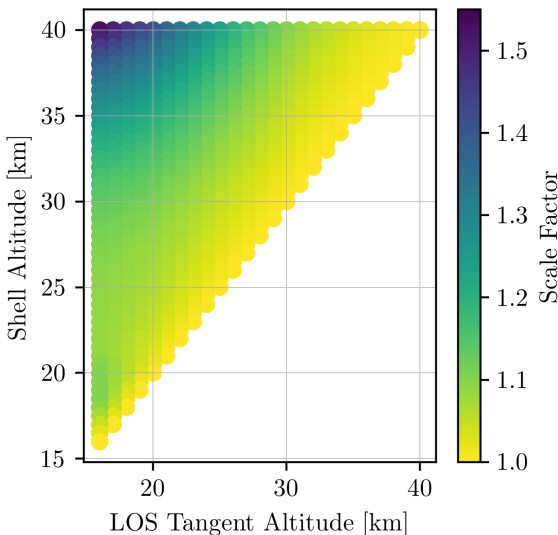

**Figure 4.** Scale factor matrix for a sunset occultation at the equator in March, simulated with PRATMO.

The LOS for a particular tangent altitude intersects all of the shells above it. To find the scale factors for a given LOS we first find the apparent local solar time at the midpoint of each path created by the intersection of that LOS with the shells.

PRATMO is then run with input ozone, temperature, and pressure from the SAGE III/ISS Level 2 scan data. The model $NO_2$ is computed at each calculated LST and at a SZA of 90°, corresponding to the exact time of sunrise or sunset. For each shell altitude along the LOS, the scale factor is the PRATMO $NO_2$ at that altitude (corresponding to the LST at that location) divided by the PRATMO $NO_2$ at the tangent point altitude for that LOS (the scale factor is 1 for the shell containing the tangent point). There is no scaling done above 40 km as the low amount of $NO_2$ can lead to unphysical scale factors and we want to prevent

abnormal values from influencing the results at lower altitudes.

Figure 4 shows the photochemical scale factor matrix for a simulated event. The values in the figure are not multiplied by the path lengths. The matrix is created such that the scale factors from opposite sides of the tangent point need to be added together. This results in a scale factor that is equal to one along the diagonal, corresponding to unscaled values at the tangent point, and greater than one everywhere else. The scale factors increase as the path length component of the LOS gets further

away from the tangent point.

It is also useful to look at the scale factor as a function of altitude along each LOS (Figure 5). Lower lines of sight pass through more layers of the atmosphere, resulting in greater scale factors. For lines of sight below about 30 km the change in scale factor with altitude becomes non-linear. This is because the shape of $NO_2$ cycle across the terminator changes with altitude (right panel of Figure 3). At higher altitudes the $NO_2$ increases along the whole LOS; below about 30 km the $NO_2$

starts to level out on the night-side (the curve on either side of the terminator becomes different), changing the slope of the scale factor curves in Figure 5.





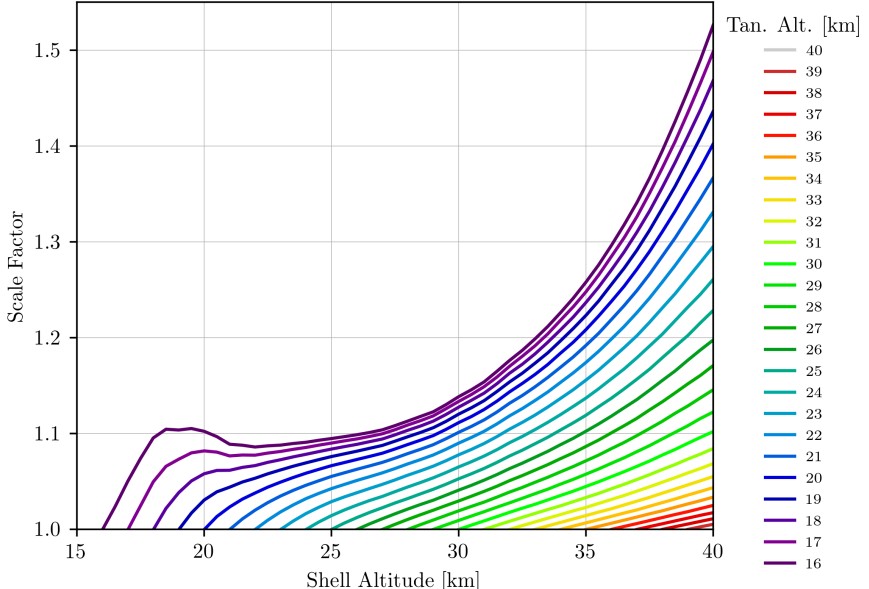

**Figure 5.** Scale factors along each line of sight for a sunset occultation at the equator in March, simulated with PRATMO.

The scaled path length matrix, $X_{dv}$, is equal to the initial path length matrix, $X_0$, where each element has been multiplied by the corresponding element from the scale factor matrix shown in Figure 4. $X_{dv}$ is used along with the calculated slant path optical depths in Equation 2 to get the $NO_2$ number density, accounting for diurnal varitations. The resulting values of $n_{dv}$

will be smaller than the original retrieved values. This is because the gradient on the near side of the LOS is smaller than the gradient on the far side, resulting in scale factors greater than one and therefore lower $NO_2$. Figure 6 shows the results for a sample SAGE III/ISS event. The left panel contains the optical depth profile, while the right panel compares the SAGE III/ISS $NO_2$ number density with the diurnally varying number density. In general the difference between the two profiles becomes greater as altitude decreases.

## 5 Results


The effect of accounting for diurnal variations on the retrieved SAGE III/ISS $NO_2$ is quantified by the difference between the SAGE v5.1 retrieval and the diurnally varying retrieval (Figure 7). In general the difference between the retrievals becomes greater than 20% below 25 km, which is larger than the reported random uncertainty in the SAGE III/ISS $NO_2$. Including the diurnal variations is more important in the winter at high latitudes; at these times the relative $NO_2$ decrease at sunrise/increase

at sunset is larger. The bias between the retrievals is not consistent between the sunrise and sunset $NO_2$ due to the differences in $NO_2$ chemistry at these times. Including diurnal variations along the LOS in the retrieval has a greater effect on sunrise than sunset above 25 km in the tropics and everywhere at higher latitudes.



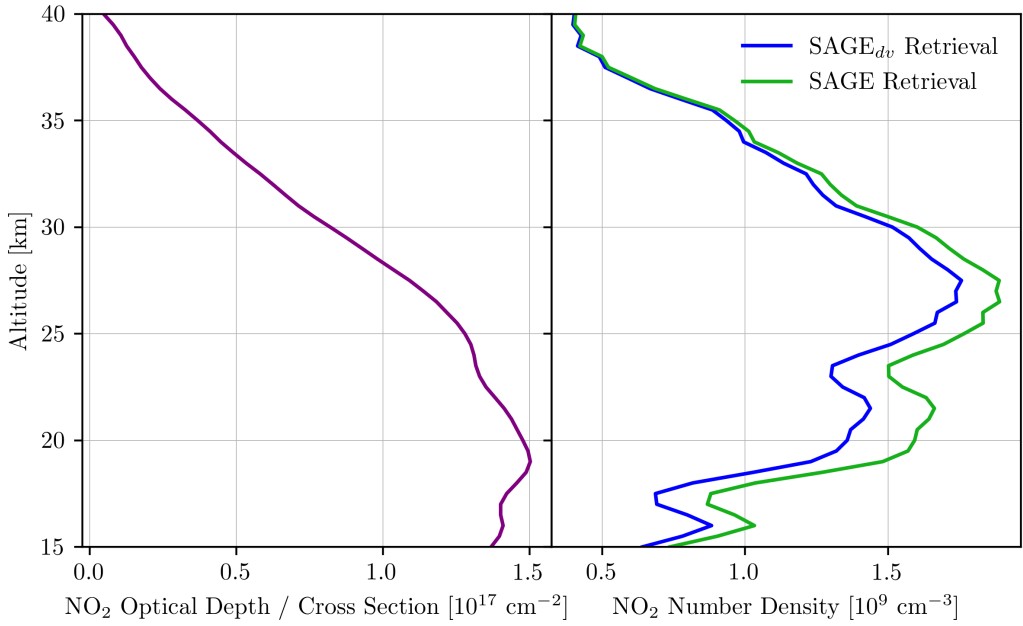

**Figure 6.** Optical depth and $NO_2$ number density for both the diurnally varying and SAGE v5.1 retrievals. Calculated for SAGE III/ISS event 967110.

The results presented in Figure 7 are very similar to those in Brohede et al. (2007), where they estimated the magnitude of neglecting diurnal variations in $NO_2$ for a simulated occultation instrument (not specific to SAGE II or III). They found that the bias increases rapidly below 25 km (below the peak in $NO_2$ density) and is larger at low latitudes for sunset. They also found that at high latitudes the bias is largest near equinoxes and sunrise values are slightly larger than sunset values. It was determined that this effect was enough to explain most of the difference between SAGE III/Meteor-3M and OSIRIS $NO_2$ at low altitudes (although they did not actually apply the correction to the SAGE III/Meteor-3M data).

The magnitude of the photochemical effect is also similar to that used in the Halogen Occultation Experiment (HALOE, Russell III et al., 1993) retrieval. HALOE is one occultation instrument that does include diurnal effects in the retrieval (Gordley et al., 1996). They use a factor based on results from the previous layer and a model that provides the $NO_2$ mixing ratio as a function of SZA and season. This is less accurate than the scale factors used in the present study, which are modelled for each $NO_2$ profile individually. The effect of the HALOE scaling is considered significant below 25 hPa ($\approx$ 27 km). They also found the diurnal effect in sunrise to be 2-3 times larger than in sunset, which is greater than the difference between sunrise and sunset observed here for SAGE III/ISS.

Accounting for diurnal variations in the retrieval changes the SAGE III/ISS $NO_2$ time series. This is shown in Figure 8 for several latitude bins at 25.5 km. The sunset $NO_2$ number density decreases by about 5% to 20%, with the largest decreases occurring in the tropics. The effect of the diurnal variations on the sunrise $NO_2$ has a more pronounced seasonal cycle than

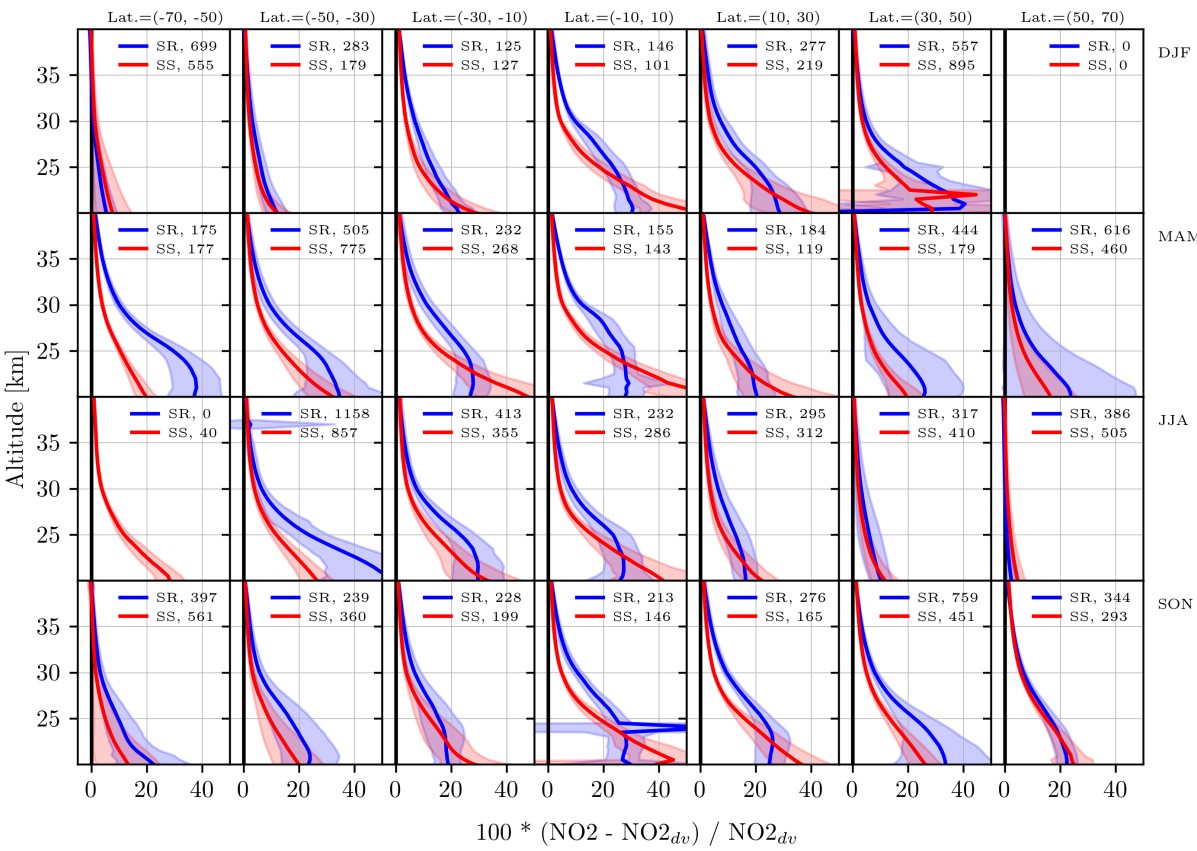

**Figure 7.** Mean difference between the SAGE v5.1 and diurnally varying (dv) retrievals for SAGE III/ISS sunset and sunrise $NO_2$. The error bars are the standard deviation.

sunset, with a greater decrease in the winter, and a difference ranging from 5% to 30%. During winter months the diurnal effect
is about 2 times greater at sunrise than at sunset, which is comparable to the difference reported for HALOE $NO_2$ in Gordley et al. (1996). However during the summer the bias is similar for both sunrise and sunset. These variations in the time series should be considered when using the SAGE III/ISS $NO_2$ data.

Both the diurnally varying and SAGE v5.1 retrievals were compared with OSIRIS $NO_2$ as a way to validate the data (panel a of Figure 9). The comparisons were done for events within 24 hours, 10° longitude, and 2° latitude, although the exact choice
of coincidence criteria has minimal effect on the results. The SAGE III/ISS profiles were shifted from sunrise and sunset to the OSIRIS measurement time using PRATMO. The SAGE III/ISS $NO_2$ is generally biased high compared to the OSIRIS $NO_2$, with a difference of up to 40% at low altitudes/high latitudes. The differences between SAGE III/ISS and OSIRIS are within the combined uncertainties of the instruments. In general the SAGE III/ISS sunrise measurements agree better with OSIRIS than sunset, which could be because OSIRIS measures close to sunrise. The comparison was also performed excluding OSIRIS

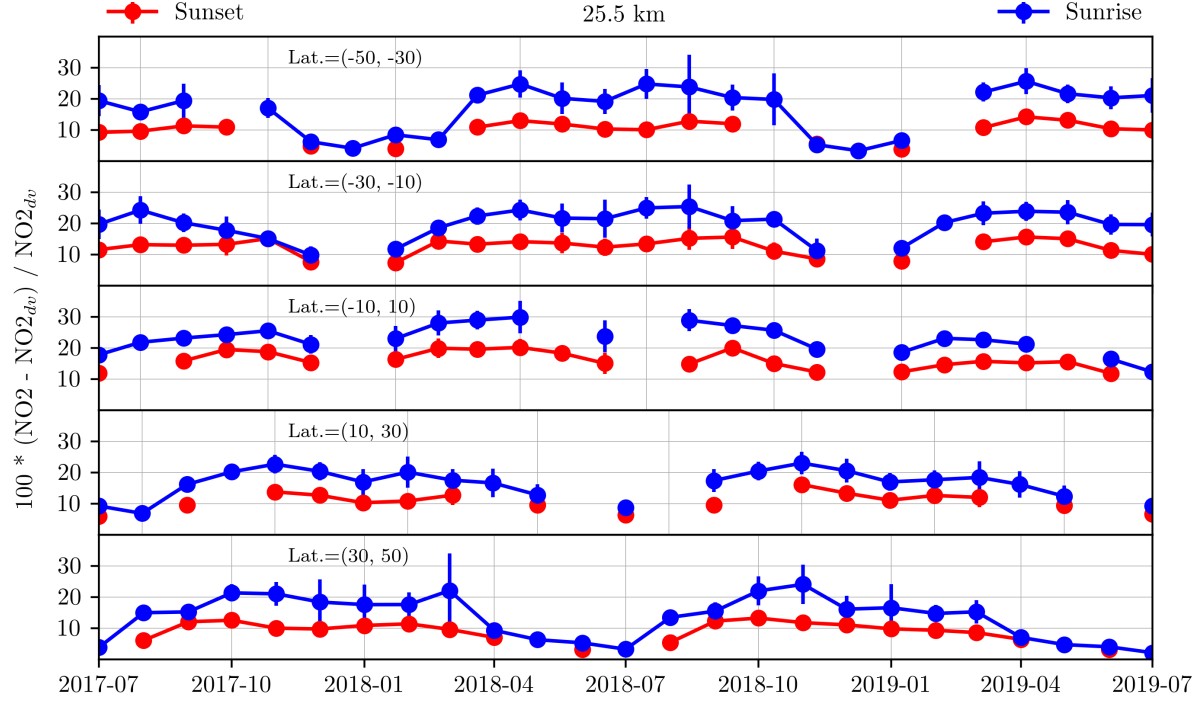

**Figure 8.** Mean difference between the SAGE v5.1 and diurnally varying ($dv$) retrievals for SAGE III/ISS sunset and sunrise $NO_2$ at 25.5 km. The error bars are the standard deviation.

profiles with a SZA angle greater than $86°$, where the diurnal effect is significant. This had a negligible effect on the difference with SAGE III/ISS.

Accounting for diurnal variations in the SAGE III/ISS retrieval improves agreement with OSIRIS by up to 20% in the midstratosphere (panel b of Figure 9). Overall the diurnally varying SAGE III/ISS $NO_2$ agrees better with the OSIRIS $NO_2$ than the SAGE v5.1 $NO_2$. The only region where this is not true is near the tropical tropopause. This area is where the diurnal effect becomes large (Figure 7), resulting in much smaller SAGE III/ISS $NO_2$ values. The sunset $NO_2$ in particular decreases by up to 50% below 25 km in the tropics. This is larger than the initial difference between SAGE III/ISS and OSIRIS, resulting in the positive bias that is observed from 15 to 25 km at low latitudes.

## 6 Conclusions

We have developed a retrieval algorithm that uses publicly available SAGE III/ISS data to account for changes in $NO_2$ along the occultation line of sight that come from the photochemically driven diurnal cycle. The retrieval relies upon scaling factors derived from a photochemical box model with input ozone, temperature, and pressure profiles taken from the reported SAGE III/ISS scan.





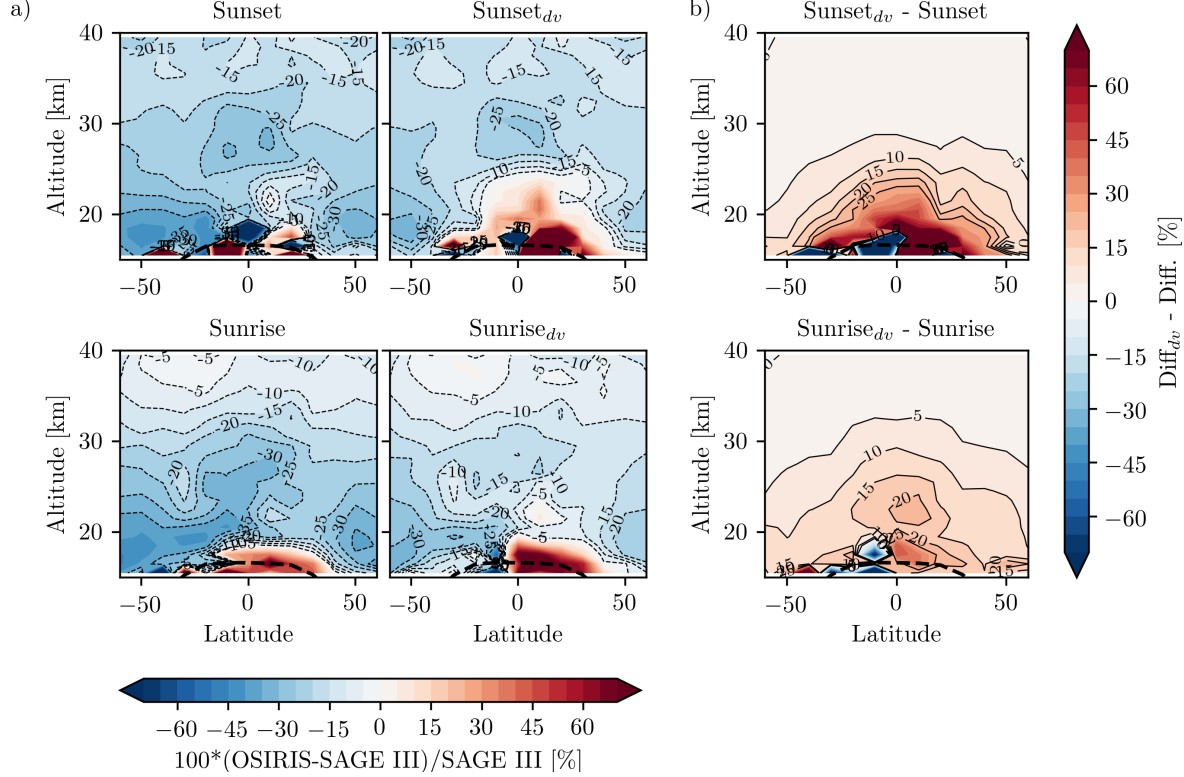

**Figure 9.** Panel a: Zonal mean percent differences of SAGE III/ISS sunset and sunrise $NO_2$ and OSIRIS $NO_2$ coincident measurements for the SAGE v5.1 and diurnally varying (*dv*) retrievals. The contour spacing is 5%. The bold dashed line is the mean tropopause location. Panel b: Difference between the columns in panel a.

It was determined that the neglect of diurnal variations in the SAGE v5.1 retrieval always biases the results high. Note that this high bias is present in $NO_2$ retrieved from any occultation instrument that neglects diurnal variability. In the case of SAGE
III/ISS $NO_2$, we found that carefully accounting for diurnal variations in the retrieval is quite important below 30 km, with an effect nearing 20% on the resulting values. The correction has the greatest effect at high winter latitudes, and is more important for sunrise occultations than sunset. These are potentially important differences in the reported $NO_2$ densities in the lower stratosphere, where several interesting chemical and dynamical science questions remain. Including diurnal variations in the $NO_2$ retrieval also has an impact on the monthly zonal mean time series which should be considered when studying long term
variability.

Accounting for these diurnal variations in the SAGE III/ISS retrieval improves the agreement between with OSIRIS $NO_2$ by up to 20% at lower altitudes. While there is a remaining bias between SAGE III/ISS and OSIRIS that is not well understood, it has a reasonable magnitude considering the very different measurement and retrieval techniques, and is within their combined uncertainties.



*Data availability.* The SAGE III/ISS Level 1 and 2 data are available through https://search.earthdata.nasa.gov/. The SAGE III/ISS $NO_2$ that was retrieved by accounting for diurnal variations is available at https://research-groups.usask.ca/osiris/data-products.php. OSIRIS $NO_2$ is also available at the previous link.

*Author contributions.* KD performed the analysis and prepared the manuscript. DZ assisted with writing the retrieval code. AB and DD proposed the original idea for the project and provided guidance throughout. RD provided assistance with using the SAGE III/ISS data. DF
and WR, along with the other authors, provided significant feedback on the analysis and the manuscript.

*Competing interests.* The authors declare that they have no conflicts of interest.

*Acknowledgements.* The authors thanks the Swedish National Space Agency and the Canadian Space Agency for the continued operation and support of Odin-OSIRIS. We also thank the SAGE III/ISS science team for providing the SAGE III/ISS data. SAGE III/ISS is a NASA Langley managed Mission funded by the NASA Science Mission Directorate within the Earth Systematic Mission Program. Enabling partners
are the NASA Human Exploration and Operations Mission Directorate, International Space Station Program and the European Space Agency. The National Center for Atmospheric Research is sponsored by the US National Science Foundation.



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
