# Peer review of "Accounting for the photochemical variation of stratospheric NO2 in the SAGE III/ISS solar occultation retrieval"

_Atmospheric Measurement Techniques, 2020_

## Referee Comment (RC1) · Anonymous Referee #2 · 14 Oct 2020

General Comments

The manuscript quantifies the impact of NO2 diurnal variability along the SAGE III/ISS line of sight and describes a new NO2 retrieval algorithm that accounts for this variability. The authors show that accounting for this effect leads to lower NO2 below 30km and improves the agreement with OSIRIS observations for the mid-stratosphere. This is a useful development that will benefit the community and the work fits well in the scope of AMT. The manuscript is well-written overall, but would benefit from some clarifications, as described in the specific comments.

Specific Comments

Line 8: What is "undoing" a retrieval? Reversing the algorithm to back out optical depth?

Line 102 and Fig. 2 caption: Which dashed line? There are 2. Maybe draw the SZA on the figure.

Line 146: The text states: "The values in the figure are not multiplied by the path lengths". Would we expect them to be?

Line 170: Please elaborate on how the bias is not consistent with the differences.

Fig. 6: A panel showing percent difference would be helpful as well.

Fig. 7: Is there bad data in the middle panel of the bottom row?

Fig. 9a: It appears that while the negative bias is reduced in the SAGEdv case, the positive biases at lower altitudes increase. This merits some discussion.

Fig. 9: Are the right (b) panels SAGEdv – SAGE, or (OSIRIS-SAGEdv) – (OSIRIS-SAGE)? In other words, why is "SAGEdv – SAGE" positive when it is stated that "neglect of diurnal variations in the SAGE v5.1 retrieval always biases the results high"? Figure 9 might be more intuitive if it were presented as SAGE – OSIRIS rather than OSIRIS – SAGE.

---

## Referee Comment (RC2) · Emmanuel Dekemper (Referee) · 19 Oct 2020

**1   General comments**

The manuscript presents the results of the investigation of the importance of correctly accounting for the diurnal cycle of NO2 in solar occultation experiments, in particular for the NO2 product of SAGE-III onboard ISS. A few previous papers had already studied this effect, but the present work addresses a clear case as the current SAGE-III retrieval algorithms do not achieve this level of sophistication. The method proposed in this work can easily be transposed to other past or future solar occultation experiments, such as

balloon-borne campaigns, or satellite instruments such as ALTIUS.

Overall, the manuscript is an excellent piece of work: the scope of the study is clearly described, the datasets used for testing the hypotheses are well chosen, the figures are all appropriate, and of very good graphical quality, and the text is in general very clear. I only have a couple of questions, and spotted a few typos.

**2 Specific comments**

1. On line 144, it is explained that the PRATMO model is not used above 40km in order to avoid unphysical values. This tends to indicate that the authors sought to use it on a larger vertical extent, but faced reliability issues. As a result, the disagreement between the two products (original SAGE-III NO2 profiles, and those corrected for diurnal variability) tends to fade away as the altitude increases (Fig.6). As a note towards readers willing to apply the same correction technique, could you elaborate a bit on the reasons which forced you to not implement the diurnal correction above 40km? Is it due to the model itself? or to the specified O3, pressure, and temperature profiles?... Was there any reliability concerns raised at lower altitudes?

2. It is not entirely clear if Fig.4, which shows the scaling factors applied to the path segment matrix X, shows an average scaling taking into account the two disymmetric sides of the light path, or if it only shows the factors for one half of the path. In the latter case, which half is it? Could you make it a bit clearer in your description?

3. On page 10, the authors point out that the agreement of the improved NO2 profiles is better at sunrise than sunset. The discussion about this difference is somewhat vague, especially that, to my knowledge, OSIRIS is sounding the atmosphere both close to local sunset and sunrise. Could this topic be slightly
expanded? For instance by listing the possible reasons for this pending bias. In particular, did you consider to restrict the coincidence criteria to less than 24 hours, in order to limit the time gap which needs to be solved by the PRATMO model?

**3  Technical corrections**

4. Line 52: double "the"

5. Line 221: "... between with ..." should read "... with ..."

---

## Author Response (AR1)

Response to Reviewers: Accounting for the photochemical variation of stratospheric NO2 in in the SAGE III/ISS solar occultation retrieval

Emmanuel Dekemper

**Specific comments**
1. On line 144, it is explained that the PRATMO model is not used above 40km in order to avoid unphysical values. This tends to indicate that the authors sought to use it on a larger vertical extent, but faced reliability issues. As a result, the disagreement between the two products (original SAGE-III NO2 profiles, and those corrected for diurnal variability) tends to fade away as the altitude increases (Fig.6). As a note towards readers willing to apply the same correction technique, could you elaborate a bit on the reasons which forced you to not implement the diurnal correction above 40km? Is it due to the model itself? or to the specified O3, pressure, and temperature profiles?... Was there any reliability concerns raised at lower altitudes?

The uncertainty in the SAGE measurements becomes greater than 20% above 40 km (and below 10 km) so we did not want to focus our results on these regions. In addition, the NO2 concentration gets very low above 40 km. When calculating the scale factors we divide by the NO2 at the tangent point altitude. For tangent points above ~40 km this results in dividing by a small number, which produces a large scale factor, despite the fact that the absolute difference in NO2 might not be so large. This has been clarified in the manuscript.

2. It is not entirely clear if Fig.4, which shows the scaling factors applied to the path segment matrix X, shows an average scaling taking into account the two disymmetric sides of the light path, or if it only shows the factors for one half of the path. In the latter case, which half is it? Could you make it a bit clearer in your description?

This figure shows the scale factors from either side of the tangent point added together. This information has been added to the figure caption.

3. On page 10, the authors point out that the agreement of the improved NO2 profiles is better at sunrise than sunset. The discussion about this difference is somewhat vague, especially that, to my knowledge, OSIRIS is sounding the atmosphere both close to local sunset and sunrise. Could this topic be slightly expanded? For instance by listing the possible reasons for this pending bias. In particular, did you consider to restrict the coincidence criteria to less than 24 hours, in order to limit the time gap which needs to be solved by the PRATMO model?

We only use the morning OSIRIS measurements (ascending node). A drift in the OSIRIS orbit resulted in many of the descending measurements occurring at night, when OSIRIS cannot measure, which affects the sampling. Because of this, it is common to use only the morning OSIRIS data. Using coincidence criteria of less than 24 hours did not result in large differences in the comparisons so we chose to have more data points in each bin. This has been clarified in the manuscript.

**Technical corrections**
4. Line 52: double "the"  Fixed
5. Line 221: "... between with ..." should read "... with ..."  Fixed

Referee #2

**Specific comments**
Line 8: What is "undoing" a retrieval? Reversing the algorithm to back out optical depth?

Yes, by undoing a retrieval we mean converting the number densities back to optical depths. This has been clarified in the text.

Line 102 and Fig. 2 caption: Which dashed line? There are 2. Maybe draw the SZA on the figure.

Both dashed lines. More detail has been added to the Figure and the text so that this is better explained.

Line 146: The text states: "The values in the figure are not multiplied by the path lengths". Would we expect them to be?

The matrix used in the retrieval generally consists only of path length elements. For the diurnally varying retrieval we are multiplying each path length by a corresponding scale factor. The figure only shows these scale factor values, as opposed to the final matrix (including path lengths) that is used to do the retrieval. So we thought it useful to clarify that this Figure only considers the scale factors that go into the final path length matrix.

Line 170: Please elaborate on how the bias is not consistent with the differences.

Figure 1 shows that the shape of the diurnal cycle across the terminator is different at sunrise and sunset, which results in different photochemical scale factors. A comment has been added to the manuscript.

Fig. 6: A panel showing percent difference would be helpful as well.

A panel showing the percent difference has been added to the figure.

Fig. 7: Is there bad data in the middle panel of the bottom row?

Yes, there were a few bad data points. The figure has been changed to only include NO2 values within five standard deviations of the mean.

Fig. 9a: It appears that while the negative bias is reduced in the SAGEdv case, the positive biases at lower altitudes increase. This merits some discussion.

This is discussed in lines 204-207 (210-213 in updated manuscript). The positive biases increase because the diurnal effect is very large near the tropical tropopause and the absolute NO2 values are low, resulting in a decrease in the diurnally varying SAGE III/ISS NO2 that is greater than the initial difference between SAGE III/ISS and OSIRIS NO2.

Fig. 9: Are the right (b) panels SAGEdv – SAGE, or (OSIRIS-SAGEdv) – (OSIRISSAGE)?
In other words, why is "SAGEdv – SAGE" positive when it is stated that "neglect of diurnal variations in the SAGE v5.1 retrieval always biases the results high"? Figure 9 might be more intuitive if it were presented as SAGE – OSIRIS rather than OSIRIS – SAGE.

The right panels are indeed the difference (OSIRIS – SAGEdv) – (OSIRIS – SAGE). This has been clarified in the figure. The figure has also been changed to show the difference SAGE – OSIRIS instead of OSIRIS – SAGE.

[revised manuscript text omitted]

---

## Author Response (AR2)

Response to Associate Editor: Accounting for the photochemical variation of stratospheric NO2 in in the SAGE III/ISS solar occultation retrieval

Thank you very much for the feedback. References to the suggested papers on ground-based NO2 retrievals have been added to the introduction of the manuscript.